# Reconstructive Surgery of Pressure Injuries in Spinal Cord Injury/Disorder Patients: Retrospective Observational Study and Proposal of an Algorithm for the Flap Choice

**DOI:** 10.3390/healthcare12010034

**Published:** 2023-12-22

**Authors:** Rossella Sgarzani, Paola Rucci, Siriana Landi, Micaela Battilana, Rita Capirossi, Beatrice Aramini, Luca Negosanti

**Affiliations:** 1Dipartimento di Scienze Mediche e Chirurgiche (DIMEC Dpt.), Bologna University, Via G. Massarenti 9, 40138 Bologna, BO, Italy; beatrice.aramini2@unibo.it; 2Dipartimento di Scienze Biomediche e Neuromotorie (DIBINEM Dpt.), Bologna University, Via Ugo Foscolo 7, 40126 Bologna, BO, Italy; paola.rucci2@unibo.it; 3Montecatone Rehabilitation Institute, Via Montecatone 37, 40026 Imola, BO, Italy; siriana.landi@montecatone.com (S.L.); micaela.battilana@montecatone.com (M.B.); rita.capirossi@montecatone.com (R.C.); luca.negosanti@montecatone.com (L.N.)

**Keywords:** spinal cord injury, pressure injury, local flap, reconstructive algorithm

## Abstract

Pressure injuries (PIs) are a common complication in patients with spinal cord injury/disorder (SCI/D), and deep PIs require surgical treatment consisting of wide debridement and adequate reconstruction. We conducted a retrospective observational study at a tertiary rehabilitation hospital for SCI/D in Italy with the aim of describing the incidence and associated risk factors of postoperative complications in individuals with SCI/D presenting with chronic deep PIs, treated with a specific flap selection algorithm based on the site of the defect, the presence of scars from previous surgeries, and the need to spare reconstructive options for possible future recurrences. Medical records of surgical procedures performed on SCI/D patients with fourth-degree PIs, according to NPUAP classification (National Pressure Ulcer Advisory Panel), between July 2011 and January 2018 were reviewed. A total of 434 surgical procedures for fourth-degree PIs in 375 SCI/D patients were analyzed. After a mean follow-up of 21 months (range 12–36), 59 PIs (13.6%) had minor complications, and 17 (3.9%) had major complications requiring reoperation. The sacral site and muscular and musculocutaneous flaps were significant risk factors for postoperative complications. Six patients (1.4%) had a recurrence. The choice of flap correlates with the outcome of decubitus reconstruction. Therefore, reconstructive planning should be based on established principles.

## 1. Introduction

Spinal cord injury/disorder (SCI/D) patients tend to develop pressure injuries (PIs) due to the lack of sensitivity in the areas where the body weight lies while the patient is in the sitting position or in bed. Continuous weight on these areas decreases blood flow to the skin and underlying tissues, ultimately causing tissue necrosis and the formation of an open wound [1], with a lifetime incidence of PIs reaching 86% in paraplegic patients [2]. Moreover, patients with SCI/D tend to develop infection due to their catabolic state; prompt wound management treatment and reduction of the sitting time are critical to prevent the evolution to deep PIs and limit infectious complications [2]. 

On the other hand, social life in SCI/D patients is interrupted during wound care treatments; therefore, a surgical approach should be offered not only to deep PIs but also to non-healing wounds.

Surgical treatment consists of wide debridement and reconstruction with local flaps. In PIs surgery, multiple factors are important to consider in order to obtain a permanently healed wound, such as a proper debridement, multiple biopsies, prolonged target antibiotic therapy, a successful reconstructive technique, and proper postoperative management [1,3]. While choosing a reconstructive flap, the surgeon should keep in mind that a SCI/D patient during his/her life may experience a local recurrence. The reported recurrence rates in specific surgical protocols are approximately 30%, and the rate of new sores is approximately 20% [4,5]. Therefore, it is mandatory not only to choose a reliable reconstructive option but also to keep a backup plan for the future.

Controversial and diverging hypotheses have been proposed on the best flap to choose, as well as the timing (one-time or two-time surgery) [6]. 

In this study, we propose a flap choice algorithm based on the site of the defect, the presence of scars from previous surgeries, and the need to spare reconstructive options for eventual future recurrence. The main aim of the work is to describe the incidence, the associated risk factors of post-surgical complications, and the recurrence in individuals with SCI/D presenting with deep PIs that are treated with the described flap choice algorithm.

## 2. Materials and Methods

We present a retrospective observational study on SCI/D adult patients affected by chronic deep PIs and treated surgically with a reconstructive flap between July 2011 and January 2018 at the Montecatone Rehabilitation Institute (a tertiary rehabilitation hospital for SCI/D in Imola, Italy). All the patients underwent wide debridement and reconstruction with local flaps, according to a specific flap choice algorithm, as described in Figure 1. 

We excluded patients with associated cerebral lesions, PIs with acute infection, and PIs with a recurrence that had already been treated surgically and included in the study.

Once the patient was enrolled in the study, data were collected from the patient from his/her medical charts during surgical admission and the 6-month and 12-month follow-up visits.

For each patient, we collected demographic data (age, sex), information about SCI/D (etiology, neurological level, and completeness according to ASIA—American Spinal Injury Association-scale), comorbidities (BMI > 30 kg/m^2^, diabetes, active smoking of more than 5 cigarettes a day, chronic kidney disease (CKD), coronary heart disease (CHD), and obstructive sleep apnea syndrome (OSAS)).

Moreover, we collected information about the PIs site (sacral, ischiatic, trochanteric, or other site) and histological diagnosis of osteomyelitis (yes/no).

Finally, we collected data on the surgical procedure: The type of flap used for the reconstruction (fasciocutaneous, fasciocutaneous island perforator flap, muscular and musculocutaneous), minor and major post-surgical complications (rated with the Clavien-Dindo grade [7], which was considered minor when rated <3 and major when rated ≥3), and the recurrence at the 6-month and 12-month follow-up visit. The study was approved by the Institutional Ethical Committee AVEC (Area Vasta Emilia Centro) with the reference number 1000-2020-OSS-AUSLIM on 19 November 2020. 

### 2.1. Statistical Analysis

Age was summarized as the mean and standard deviation (SD) and compared between groups using a one-way ANOVA. Categorical variables were described using frequencies and percentages (%), and their association with the outcomes was investigated using the Chi-squared test or Fisher’s exact test when appropriate. Variables showing significant associations at *p* < 0.1 were further investigated in logistic regression models. Specifically, multinomial logistic regression was used to predict complications (coded as no complications, minor, and major complications), and binary logistic regression was used to predict recurrence (coded as present or absent). 

In the analyses concerning the PIs, the statistical unit of analysis was the PIs and not the patient. Indeed, during the study period, a patient could have undergone more than one surgical treatment for one or more PIs. Robust standard errors were calculated to account for possible multiple interventions, i.e., dependency among observations. Incidence with a 95% confidence interval (95% CI) was reported for minor complications, major complications, and the occurrence of any complications regardless of the degree of severity and recurrence. Regression estimates were reported as Risk Ratios (RR) and 95% CI. All analyses were performed using Stata statistical software version 18 (StataCorp. 2017. *Stata Statistical Software: Release 15*. College Station, TX: StataCorp LLC), and the significance level was set to *p* < 0.05.

### 2.2. Surgical Treatment and Reconstructive Flap Choice Algorithm

The patient was operated on by a plastic surgeon in collaboration, if needed, with other surgical specialists (a urologist, general surgeon, orthopedic, and gynecologist) as in cases of PIs extending to the enteric or urinary tract. We do not use any pre-operative imaging method to diagnose osteomyelitis due to the poor predictive value [8] and prefer to treat all the patients as if they were affected by osteomyelitis. In fact, the surgical treatment always starts with the use of methylene blue to color the wound, and then a wide debridement of all colored soft and bony tissues is performed. Multiple specimens of bony tissues are always sent to pathology from the operating room to allow histological diagnosis of osteomyelitis. Histology of soft tissues is always sent to pathology from the operating room to rule out the diagnosis of a Marjolin ulcer [9]. None of the patients received any antimicrobial therapy in the 20 days before the operation, and multiple samples of soft and bony tissues are always sent to microbiology from the operating room to allow the administration of the correct antimicrobial therapy, as pre-operative swabs are not reliable [10,11].

The reconstructive procedure aims to close the wound with vital tissues, fill in all the dead space, and bring some thickness to weight-bearing areas. Keeping in mind that a recurrence may happen in the future, flap donor sites must be selected carefully, sparing further options. As an example in sacral PI, we categorically avoid the use of bilateral V to Y musculo-cutaneous advancement flaps that would not leave any option as a donor site for future reintervention. A unilateral fasciocutaneous flap is the preferred choice, leaving for eventual future needs the contralateral donor site and the gluteus maximus flap bilaterally. Our reconstructive flap choice algorithm is presented in Figure 1.

In trochanteric PIs, the debridement often produces wide defects, especially if the resection is extended to the femoris head and the acetabulum. Therefore, the reconstruction is generally delayed after one month of topical negative pressure therapy to reduce wound dimensions. Delayed reconstruction of trochanteric PIs aims to fill in the dead space. Therefore we prefer a muscle or musculo-cutaneous flap harvested from the thigh (rectus femoris muscle flap or vastus lateralis musculo-cutaneous flap, tensor fasciae latae musculo-cutaneous flap). When a muscle flap is used to fill in the dead space, a minimal skin rotation is used to close the skin layer over the flap. In spastic SCI/D patients, we usually treat the chosen muscle with botulinum toxin 10 days before surgery in order to reduce the risks related to spasms after reconstruction.

In sacral, ischial, or other PIs, the reconstruction is usually performed in one stage after the debridement. The reconstruction is usually based on local fascio-cutaneous flaps (rotation, transposition, or island perforator flaps). All procedures were performed by senior plastic surgeons trained in perforator surgery, but we employed island perforator flaps only if, during the dissection of a fasciocutaneous flap, a major perforator is directly visualized. Part of the flap is usually deepithelized in order to fill in the dead space and increase the thickness of weight-bearing areas (Figure 2).

In sacral PIs, we always perform a unilateral flap for the closure in order to spare a donor site for eventual future recurrence and reintervention.

In case of recurrence or scars from previous surgeries, muscle flaps may also be considered for sacral and ischial PIs (gluteus maximus muscle flap, semitendinosus muscle flap, and semimembranosus muscle flap). As for the trochanteric region, when a muscle flap is used to fill in the dead space, a minimal skin rotation is used to close the skin layer over the flap. 

After reconstruction, suction drains are always placed in every site except the malleolar region. Drains are usually removed after 5–7 days. 

In malleolar PIs, osteomyelitis is usually reported, and we prefer to widely resect the bone in order to perform a tension-free fasciocutaneous rotation flap.

Postoperative treatment includes the administration of piperacillin-tazobactam (18 g daily) until antimicrobial therapy can be adjusted according to the results of the specimen culture. The patient lays in bed for three postoperative weeks, avoiding laying on the flap. Every three hours, the patient’s position is changed by our operators to prevent secondary sores; the patient is not allowed to change his position in autonomy to prevent any friction or traumatism. When multiple wounds are treated in a single stage, we prefer to position the patient on a fluidized bed. Every two days, the wound is dressed by a wound care nurse. Twice a week, the wound is examined by the plastic surgeon to provide prompt treatment of the eventual hematomas, seromas, or dehiscence. Antimicrobial therapy is adjusted by the infectious disease specialist based on the histological and microbiological findings on surgical specimens. After three weeks, the stitches are removed, and the patient starts his or her rehabilitation to the sitting position. Our protocol consists of a gradual sitting training: 1 h maximum of sitting daily for 1 week, then 2 h maximum of sitting for 2 weeks [12]. 

## 3. Results

### Patients’ Characteristics and Association with Minor and Major Complications

A total of 375 patients were included in the study. They were 87.5% male and had a mean age of 49.8 (SD = 13.9) years, ranging from 15 to 80 years. The aetiology of SCI/D was traumatic in 86.9% of patients, vascular in (8%), or iatrogenic/other in 5.1%. Tetraplegia and completeness of lesion (AIS A) were found in 27.5% and 84.5% of patients, respectively. The average length of stay in the rehabilitation hospital was 55 days (ranging from 42 to 150).

A total of 44 patients underwent more than one surgical procedure (at different sites of PIs) during the study period, and 71 patients underwent multiple PI treatments at the same surgical procedure. 

A total of 434 PIs were treated surgically, with 59 PIs (in 46 patients) presenting with postoperative minor dehiscence that was classified as a minor complication, grade 1–2, according to Clavien-Dindo [7]. All minor dehiscence was treated conservatively until complete healing. 

In 17 PIs (15 patients), a major complication occurred, requiring re-intervention. Major complications were partial flap necrosis (n = 5), chronic seroma (n = 4), fistula (n = 6), and hematoma (n = 2). 

The incidence of major, minor, and all complications was 3.9% (2.3–6.2%), 13.6% (95% CI 10.5–17.2), and 17.5% (95% CI 13.8–21.2%), respectively.

Correlations between complications and patients’ demographic characteristics, level, completeness of SCI/D, and aetiology of SCI/D are reported in Table 1. 

Complications were unrelated to age, gender, tetraplegia, and AIS level. However, patients with traumatic etiology were significantly less likely to experience major complications.

The treated PIs were ischiatic in 56.2% of the cases (n = 244), sacral in 32.5% of the cases (n = 141), trochanteric in 15.7% of the cases (n = 68), and 5.8% were in other sites (n = 25). Table 2 shows that in sacral PIs, minor complications occurred in a significantly higher percentage compared to the other sites (22%), while the reverse was true for ischiatic PIs that had significantly lower rates of minor complications (9.8%). Major complications occurred with similar rates across sacral, ischiatic, and trochanter sites. 

Reconstructive flaps were fasciocutaneous in 347 PIs (80%), fasciocutaneous island perforator flaps in 14 cases (3.2%), and muscular or musculocutaneous in 74 PIs (16.8%). Table 2 shows that although the type of reconstructive flap was not statistically associated with complications, no minor or major complication was found in the PIs treated with fasciocutaneous island perforator flap.

In 239 cases (55.1%), a histological diagnosis of osteomyelitis was confirmed. This diagnosis did not confer a higher risk of complications. Treatment of multiple PIs in one stage or in different sites was unrelated to complications. Notably, no major complication was found when treatment included a fluidized bed. 

A multinomial regression model was then carried out to predict minor and major complications as a function of the three patient and PI variables identified as significant, i.e., sacral site, ischial site, and etiology. Sacral PIs were associated with a two-fold risk of minor complications (RR = 2.27, 95% CI 1.24–4.13), ischial PIs were unrelated with minor and major complications (RR = 0.70, 95% CI 0.36–1.33), and vascular etiology had a five-fold risk of major complications (RR = 5.05, 95% 1.47–17.32, Table 3).

After omitting the ischial site in a simplified model, sacral PIs were associated with a two-fold risk of minor complications (RR = 2.69, 95% CI 1.55–4.67), and vascular etiology was associated with a five-fold risk of major complications (RR = 5.01, 95% 1.47–17.05).

All the enrolled patients were followed up at 6 and 12 months, but for many of them, we also have data afterward, as the patients are followed chronically at our center for multiple conditions connected to spinal cord injury, and we keep on collecting data on operated sites at every visit. After a mean follow-up of 21 months (ranging from 12 to 36), a recurrence was found in 6 patients (1.4%). Neither the level and cause of SCI nor age and sex were associated with the recurrence (Table 4). All patients with a recurrence had at least one comorbidity vs. 56.6% without a recurrence (*p* = 0.033). Specifically, 4 were active smokers, 1 of whom also had chronic kidney disease and chronic heart disease; 1 had obesity and chronic kidney disease; 1 patient was obese; and another had diabetes. Notably, in all recurrences, a fasciocutaneous reconstructive flap was used.

## 4. Discussion

PIs are a common complication in SCI/D patients. Despite all the efforts spent on preventive strategies, the use of anti-decubitus surfaces, and patients’ training, the lifetime incidence of PI reaches 86% in paraplegic patients [2]. In the case of deep PIs, classified as grade III or IV according to the NPUAP classification (National Pressure Ulcer Advisory Panel) [13], surgical treatment must be considered. It is universally accepted that surgical treatment consists of debridement and reconstruction with flaps. On the other hand, there is still debate on the main key factors, which are flap choice and postoperative care (short stay in a surgical ward or long stay in a rehabilitation ward).

In 2016, Tadiparthi proposed a multidisciplinary approach [14]; we believe that a clinical pathway with specific pre- and postoperative workup is fundamental [14,15,16,17]. Contrary to Tadiparthi, we prefer to hospitalize the patient until the wound is completely healed [12] and rehabilitate the patient to the sitting position to prevent immediate complications and the tendency of recurrence. We realize that the costs of a long hospitalization in a rehabilitation institute are relevant. On the other hand, it is difficult to quantify how much we save by preventing new pressure sore formation with this approach [18]. Although recent publications have tried to delineate evidence-based treatment guidelines, they often lack decision-making algorithms for selecting a flap according to the size and location of the ulcer. Furthermore, none of them take into consideration the presence of scars from previous surgeries [19]. 

In the literature, we found multiple reports on the surgical approach and reconstructive choices for SCI patients with pressure sores [3,4,5,20,21,22]. The reported complication rates are very heterogeneous. Disa et al. reported complication rates of 31% following pressure ulcer reconstruction with mixed flap compositions [20]. Mandrekas and Mastorakos reported complication rates as low as 7% in the use of myocutaneous flaps for pressure sore reconstruction, whereas others, such as Tavakoli et al., reported complication rates as high as 62% following reconstruction with myocutaneous flaps [21,22]. In our series, the overall incidence of complications was 17.5%, where sacral site and muscular and musculo-cutaneous flaps were significant risk factors associated with post-surgical complications.

In 2011, a group of researchers from Gent, Belgium, published a study that demonstrated that including muscle in the transferred flap was not necessary because it was not superior to fasciocutaneous or perforator flaps; moreover, perforator flaps left minor residual morbidity [23]. Even before the perforator movement in 1993, Yamamoto et al. advocated the superiority of fasciocutaneous flaps in the reconstruction of sacral pressure sores that allowed important future options [24]. In 2003, Pufe described how the blood supply around pressure sores is augmented due to the expression of angiogenic factors [25]. Based on this observation, in 2017, Kelahmetoglu described propeller perforator flaps based on enlarged perforators in the chronic PI margin [26]. Balakrishnan, in 2020, proposed the use of a single best perforator-based Pacman flap (SBPBPF) to cover pressure sores of any dimension, as it fortifies the advancement and transposition flap biogeometry principles with robust blood supply of perforator flaps [27]. 

We tend to use perforator flaps only if, during the dissection of a fasciocutaneous flap, a major perforator is directly visualized. In a systematic review by Sameeem et al. [28], an overall complication rate of 19.6% is reported for perforator flaps, while fasciocutaneous and musculocutaneous flaps reported complications rates were 11.7% and 18.6%, respectively, even if a statistical significance was not demonstrated. Vathulya [29] reported an overall complications rate of 18.8% and a recurrence rate of 7.5% concerning perforator flaps, higher than that reported in our series, where perforator flaps are performed only in particularly favorable conditions.

The goals of surgical reconstruction are to cover the wound effectively with vital tissues that may bring antimicrobial therapy to the wound bed, to fill up all dead spaces, and to leave other options for any recurrence. In the literature, the use of muscle flaps is still often advocated in order to fill in the dead space. As an example, in 2009, Romy Ahluwalia et al. reported a retrospective analysis of 72 ischial pressure injuries treated with a flap, concluding that a posterior medial thigh fasciocutaneous flap combined with a biceps femoris muscle flap (the approach used only in 29 cases of their series) was their first choice in ischial pressure wound reconstruction [30]. Instead of using muscle flaps to fill in dead spaces, in 2017, Gargano et al. reported their experience with the use of a technique they called Cone of Pressure (COP) flap, a rotation fasciocutaneous flap that is partially deepithelialized and inset with transcutaneous nonabsorbable sutures, advocating that a COP flap provides padding over a bony prominence and significantly reduces recurrences [31]. We agree with this approach; in fact, part of our fasciocutaneous flaps are usually deepithelized in order to fill in the dead space and increase the thickness of weight-bearing areas.

Already in 1986, Kauer et al. proposed flap selection based on the need for skin and muscle-saving techniques because of the likelihood of recurrent or multiple wounds amongst spinal cord injury patients [32]. In fact, we have to adequately consider the risk of recurrence in these patients, and we must spare donor sets for further easy reconstructive options. To achieve these results, we usually consider, in the case of ischial or sacral PIs, fasciocutaneous flaps as the first choice to leave muscular flaps as a second choice for eventual recurrence. In the case of sacral PI, the flap is always harvested unilaterally for the same reason. In 2017, MJ Alfeehan et al. published their experience with muscle- and fascia-sparing random pattern hatchet flaps in the primary reconstruction of pressure sores in different body regions, showing it is a reliable option that does not compromise options for future repairs [33].

In our experience, only in the case of trochanteric PIs is muscular flap reconstruction (rectus femoris or lateral vastus muscular or myocutaneous flap) preferred to fill in all dead space. In fact, we agree with Daneshgaran et al. [34] that if bone infection is present, then the Girdlestone procedure, which consists of femoral head osteotomy followed by muscle flap closure of the resulting defect, has demonstrated benefits in spinal cord injury patients. This kind of resection is wide and results in a large and deep defect that is difficult to completely fill with a fasciocutaneous flap. In this region, a fasciocutaneous flap can adequately cover the surface but might leave deep dead space that can degenerate into a hematoma or chronic seroma. We usually administer topical negative pressure therapy after debridement in order to reduce the dead space and prepare the site for a secondary reconstruction that is adequately performed with a muscle or miocutanous flap [35]. This two-time intervention represents, in our experience, a good choice to reduce complications and allow a good and permanent reconstruction. We must consider that in SCI/D patients, a muscle from the thigh can be easily used for reconstruction without any functional mobility in the donor site, so the rectus femoris or vastus lateralis lap can be harvested. We underline that SCI/D patients are frequently affected by spasticity, and it must be taken into consideration. A spasm after surgery can lead to muscle flap or stitches detachment and failure of the reconstruction. In these cases, the evaluation must be made together with a specialist in spasticity treatment in order to minimize postoperative complications. In fact, the patient is usually evaluated by a plastic surgeon and spasticity specialist, and the chosen muscle is treated with botulinum toxin before surgery. We usually treat each muscle indicated by the plastic surgeon with botulinum toxin 10 days before surgery in order to have the effect after the surgical procedure. An injection of 100 UI of botulinum toxin is injected at two points into the muscle under an ultrasound guide.

This last aspect is important to underline that a multidisciplinary approach is fundamental in this patient, that have a lot of specific aspects to consider. The team dedicated to the SCI/D patient affected by PIs in our hospital includes a plastic surgeon, nurses specialized in wound care, a physiatrist, a physiotherapist, an orthopedic surgeon, a general surgeon, a urologist, a neurophysiologist and, if necessary, other specialists can be considered, such as a vertebral surgeon or an oncologist. 

The recurrence rate reported in the study (1.4%) in 6 patients after a mean follow-up of 21 months is satisfactory compared to other reported results [29] and depends on very accurate management. All patients with a recurrence had at least one comorbid condition. A correlation between recurrence and comorbidities was already reported by Bamba et al. [36], who found that BMI < 18.5 and active smoking were independent risk factors for pressure ulcer recurrence.

The findings of this study may have practical implications for clinical practice, as the satisfactory results achieved with the described flap choice algorithm suggest it should be integrated into existing treatment protocols.

The study presents some limitations, such as the retrospective nature. Moreover, we are using fasciocutaneous flaps in non-scarred surgical sites and muscular flaps in scarred surgical sites, which can be considered a bias for the results. On the other hand, when scars are present at the surgical site, skin irritation may be jeopardized. Therefore, any fasciocutaneous flap is at risk of failure, while a musculocutaneous flap with deep vascularization from muscular vessels is safer.

## 5. Conclusions

Flap choice and PI site correlate with the outcomes of PI reconstruction. Therefore, the reconstructive planning should be based on confirmed principles. The necessity to reduce the risk of complications may include filling any dead space under the wound, having tension-free sutures, and adequately providing soft tissues over the bone prominences. Furthermore, reconstructive options need to be spared for eventual future recurrence. A multidisciplinary approach is fundamental in SCI/D patients, involving all the possible professions that can treat all the specific aspects of this complex pathology. All the aspects must be taken into consideration to have the highest possible results in this kind of surgery. Further prospective studies should be implemented to compare flap choice with the results.

## Figures and Tables

**Figure 1 healthcare-12-00034-f001:**
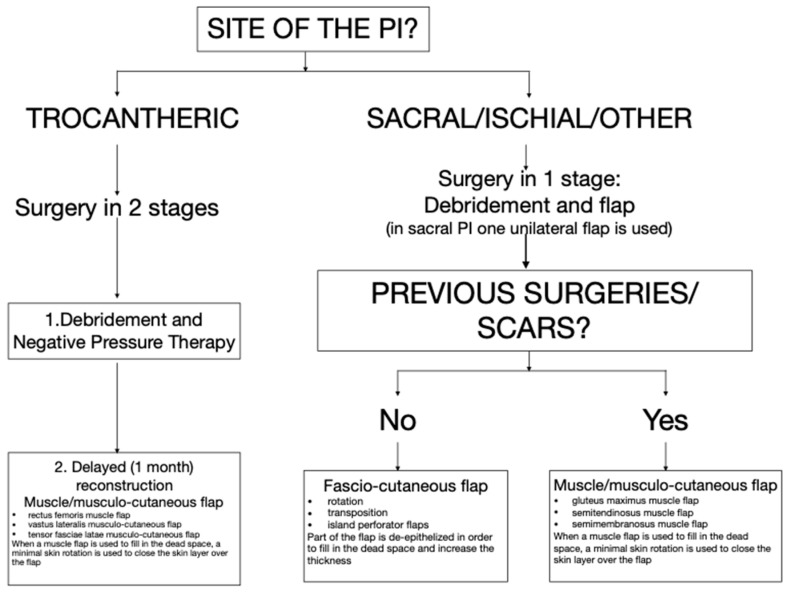
The algorithm for the reconstructive flap choice for PIs in SCI/D patients.

**Figure 2 healthcare-12-00034-f002:**
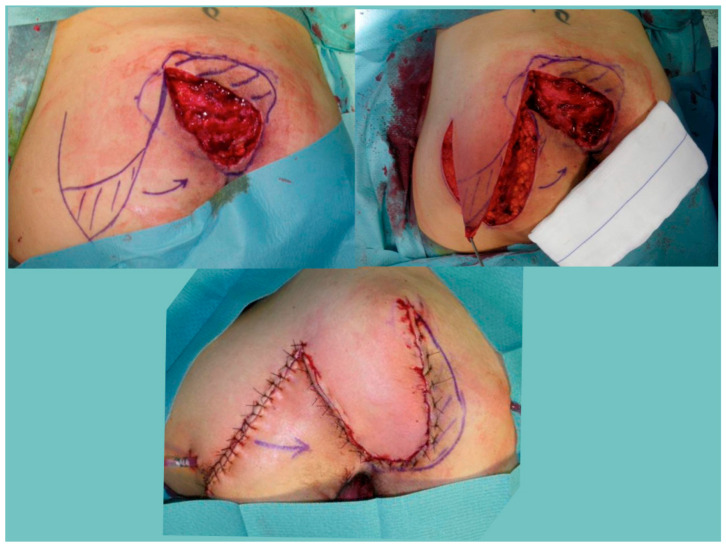
The transposition of a unilateral fasciocutaneous flap to repair sacral PIs. Part of the flap is deepithelized in order to fill in the dead space and increase the thickness of weight-bearing areas.

**Table 1 healthcare-12-00034-t001:** The characteristics of the overall sample and comparison among patients without post-surgical complications and with minor and major complications.

Pre-Surgical Characteristics	Total (n = 375)	Without Complications (n = 314)	With Minor Complications (n = 46)	With Major Complications (n = 15)	*p*-Value
Age, mean ± SD	49.8 ± 13.9	49.6 ± 14.1	50.9 ± 12.9	50.1 ± 13.3	0.821
Gender M, n (%)	328 (87.5)	274 (87.3)	39 (84.8)	15 (100)	0.292
Aetiology, n (%)					0.034 *
Trauma	326 (86.9)	277 (84.9)	40 (12.3)	9 (2.8)	
Vascular	30 (8.0)	22 (73.4)	4 (13.3)	4 (13.3)	
Other	19 (5.1)	15 (78.9)	2 (10.5)	2 (10.5)	
Tetraplegic, n (%)	103 (27.5)	90 (28.7)	10 (21.7)	3 (20.0)	0.496
Complete lesion (AIS A), n (%)	317 (85.7)	264 (84.1)	39 (84.8)	14 (93.3)	0.625
At least one comorbidity, n (%)	214 (57.1)	176 (56.1)	31 (67.4)	7 (46.7)	0.247
Diabetes, n (%)	42 (11.2)	35 (11.1)	4 (8.7)	3 (20.0)	0.482
Obesity, n (%)	103 (27.5)	86 (27.4)	14 (30.4)	3 (20.0)	0.732
Smoking, n (%)	114 (30.4)	92 (29.3)	18 (39.1)	4 (26.7)	0.380
CKD, n (%)	13 (3.5)	12 (3.8)	1 (2.2)	0 (0)	0.642
CHD, n (%)	8 (2.1)	7 (2.2)	1 (2.2)	0 (0)	0.843
OSAS, n (%)	6 (1.6)	5 (1.6)	0 (0)	1 (6.7)	0.203

* *p*-value based on the 3 × 3 table.

**Table 2 healthcare-12-00034-t002:** The characteristics of PI and comparison among PI without post-surgical complications and with minor and major complications.

PI and Surgical Characteristics	Total (n = 434)	Without Complications (n = 358)	With Minor Complications (n = 59)	With Major Complications (n = 17)	*p*-Value
More than 1 PI treated in one stage	71 (16.4)	59 (16.5)	10 (16.9)	2 (11.8)	0.869
Sacral, n (%)	141 (32.5)	105 (74.5)	31 (22.0)	5 (3.5)	0.002
Ischiatic, n (%)	244 (56.2)	210 (86.1)	24 (9.8)	10 (4.1)	0.035
Trochanter, n (%)	68 (15.7)	54 (79.4)	11 (16.2)	3 (4.4)	0.764
Other site, n (%)	25 (5.8)	24 (96.0)	1 (4.0)	0 (0)	0.181
More than 1 PI treated (different sites)	44 (10.1)	35 (9.8)	8 (13.6)	1 (5.9)	0.563
Reconstructive flap, n (%)					0.280 *
Fasciocutaneous	347 (80.0)	288 (80.4)	45 (76.3)	14 (82.4)	
Muscular or musculocutaneous	73 (16.8)	56 (15.6)	14 (23.7)	3 (17.6)	
Fasciocutaneous Island perforator flap	14 (3.2)	14 (3.9)	0	0	
Osteomyelitis, n (%)	239 (55.1)	202 (56.4)	29 (49.2)	8 (47.1)	0.463
Fluidized bed, n (%)	60 (13.8)	49 (13.7)	11 (18.6)	0 (0)	0.143

* *p*-value based on the 3 × 3 table.

**Table 3 healthcare-12-00034-t003:** A multinomial logistic regression analysis to examine the independent factors associated with post-surgical minor and major complications.

	Full Model RR (95% CI)	*p*-Value	Simplified Model RR (95% CI)	*p*-Value
Minor complications				
Sacral PI	2.27 (1.24–4.13)	0.007	2.69 (1.55–4.67)	<0.001
Ischiatic PI	0.70 (0.36–1.33)	0.277	-	-
Vascular aetiology of the SCI/D vs. traumatic	1.21 (0.38–3.80)	0.743	1.25 (0.40–3.89)	0.705
Other aetiology vs. traumatic	0.86 (0.18–4.20)	0.852	0.88 (0.18–4.28)	0.875
Major complications				
Sacral PI	1.29 (0.37–4.46)	0.686	1.23 (0.41–3.68)	0.704
Ischiatic PI	1.08 (0.35–3.37)	0.891	-	
Vascular aetiology of the SCI/D vs. traumatic	5.05 (1.47–17.32)	0.01	5.01 (1.47–17.05)	0.01
Other aetiology vs. traumatic	3.74 (0.73–19.18)	0.114	3.72 (0.73–18.99)	0.114

**Table 4 healthcare-12-00034-t004:** The characteristics of patients with and without a recurrence.

Pre-Surgical Characteristics	With Recurrence (n = 6)	Without Recurrence (n = 369)	*p*-Value
Age, mean ± SD	45.7 ± 20.1	49.8 ± 13.8	0.234
Gender M, n (%)	6 (100)	322 (87.3)	0.350
Aetiology, n (%)			0.347 *
Trauma	5 (83.3)	321 (87.0)	
Vascular	0 (0)	30 (8.1)	
Other	1 (16.7)	18 (4.9)	
Tetraplegic, n (%)	1 (16.7)	102 (27.6)	0.550
Complete lesion (AIS A), n (%)	4 (66.7)	313 (84.8)	0.222
At least one comorbidity, n (%)	6 (100)	209 (56.6)	0.033
Diabetes, n (%)	1 (16.7)	41 (11.1)	0.669
Obesity, n (%)	2 (33.3)	101 (27.4)	0.746
Smoking, n (%)	4 (66.7)	111 (30.1)	0.054
CKD, n (%)	2 (33.3)	12 (3.3)	0.018
CHD, n (%)	1 (16.7)	8 (2.2)	0.136
OSAS, n (%)	0 (0)	6 (1.6)	0.753

* *p*-value based on the 3 × 3 table.

## Data Availability

The data presented in this study are available on request from the corresponding author. The data are not publicly available due to privacy.

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
