# Peer review of "Reconstructive Surgery of Pressure Injuries in Spinal Cord Injury/Disorder Patients: Retrospective Observational Study and Proposal of an Algorithm for the Flap Choice"

_healthcare, 2023, doi:10.3390/healthcare12010034_

Round 1
Reviewer 1 Report (Previous Reviewer 1)
Comments and Suggestions for Authors
The reviewer is satisfied with the revision, and recommends being published.
Author Response
Dear reviewer,
thank you for your time and for your positive decision about the revised manuscript.
Kind regards,
Rossella Sgarzani
Reviewer 2 Report (Previous Reviewer 2)
Comments and Suggestions for Authors
Revisions have been done properly.
Author Response
Dear reviewer,
thank you for your time and for your positive decision about the revised manuscript.
Kind regards,
Rossella Sgarzani
Reviewer 3 Report (New Reviewer)
Comments and Suggestions for Authors
Overall, this paper presents an important clinical issue and a thoughtful approach to reconstructive surgery for pressure injuries in spinal cord injury/disorder (SCI/D) patients. The authors have experience in this field and offer useful principles for flap choice and surgical planning. My feedback to the authors would be the following:
- Discussion of Limitations: While some limitations are acknowledged, a more thorough discussion would strengthen the paper. This includes limitations such as the retrospective nature of the study, potential biases/confounders, potential lack of generalisability of findings.
- Clinical Implications: More detailed discussion on the practical implications of the findings for clinical practice would be valuable. This could include how the proposed algorithm can be integrated into existing treatment protocols.
- Recommendations for Future Research: The conclusion could be expanded to include more specific recommendations for future research, which would be helpful for advancing the field.
Author Response
Dear reviewer,
thank you for your time and your usefull comments.
The paper was revised according to your suggestion:
- Discussion of Limitations: While some limitations are acknowledged, a more thorough discussion would strengthen the paper. This includes limitations such as the retrospective nature of the study, potential biases/confounders, potential lack of generalisability of findings.
The discussion of limitations was implemented in the paper.
- Clinical Implications: More detailed discussion on the practical implications of the findings for clinical practice would be valuable. This could include how the proposed algorithm can be integrated into existing treatment protocols.
A sentence on clinical implications of the findings was added in the discussion.
- Recommendations for Future Research: The conclusion could be expanded to include more specific recommendations for future research, which would be helpful for advancing the field.
A sentence on recommendations for future research was added in the conclusion.
Kind regards,
The authors
This manuscript is a resubmission of an earlier submission. The following is a list of the peer review reports and author responses from that submission.
Round 1
Reviewer 1 Report
Comments and Suggestions for Authors
Please see the attachment.

Reviewer 2 Report
Comments and Suggestions for Authors
In this study the results of the clinical pathway of a tertiary hospital for fourth degree PI in SCI/D patients are described.
The clinical pathway is rational and useful and the results and especially the complication rates are very good, there are major drawbacks of design and presentation of this study.
Abstract:
Fourth degree PI are described.
There are different classifications for PIs. It should be clarified, which one is used.
Materials and Methods
Line 73 and 74:
Data were collected of the 6 and 12 month follow-up visits. However it is described that mean follow-up was 21 month. Does this mean, that the patients were revised at that point of time? Or were the operations in the mean 21 month before the data was collected. And all data is used was at 12 month post-op. This should be clarified.
Line 77:
BMI greater 20kg/m² is named as comorbidity. Between 17.5 an 25 is mostly referred a normal weight.
Even though there is some evidence that underweight is a risk factor for PI and recurrence of PI in special patients collectives. A regression model could be interesting to find out which BMI is accompanied with higher recurrence rates.
Figure 1:
To pathways are distinguished:
1. trocantheric and 2. sacral/ischial/other
For group 2 a fasciocutaneous flap in primary operations and a muscle flap for revision operations is recommended.
However, in other are all types of PIs included. Like malleolar region.
In this region, to my knowledge, no useful muscle flap is available.
Also a one time approach is recommended. Many operators tend to delay their flaps in malleolar region to achieve higher flap survival rates.
For all of this reasons, the pathway need major revisions:
Either the “other” sides should made to a third group on their own, or these PIs should be excluded from the study and the study should focus on both remaining groups.
The surgical treatment plan sounds overall reasonable.
The botulinum toxin therapy might be interesting for the readership and should be described more precisely. What dosage? How are the injection points are chosen? How is confirmed that the right muscle is injected?
Also this should be statistically evaluated. Are there differences in complications and/ or recurrence rate in muscle groups with or without BTX?
Line 137/138
Perforator flaps are only chosen, when a major perforator is directly visualized. This sounds quite random. One could argue, that more experienced surgeons will see more perforators than unexperienced surgeons. Pre-operative workup like duplex could lead to higher rates of perforator flaps. It remains unclear why a perforator flap is chosen at all. This all should be at least discussed, or the perforator group should be excluded.
Results
P- Values are missing in Table 3 and Table 5.
Lile 215:
Muscle flaps are associated with higher complications. Recording to the pathway, muscle flaps are used if scars are present, Therefore muscle flaps are used in cases, when there where already operations were performed at this surgical side. Because of that, it remains unclear if the muscle flap or the revisionary surgery is associated with higher complication rates.
Line 218:
Recurrence rate is extremely good. However follow up is too short to allow conclusions on recurrence rate. This has to be discussed.
Discussion
Line 234:
Holistic approach: What is holistic in this approach? It is a clinical pathway. Is special pre- or post-operative work-up performed?
Line 250:
Muscle flaps are associated with higher complication rates.
This contradicts the presented pathway:
Why change the pathway from fasciocutaneous flaps to muscle flaps when scars are present, if muscle flaps have higher complication rates. To substantiate the clinical pathway two groups with scars at surgical site should be distinguished. One treated with fasciocutaneous flaps and the other group with muscle flaps. Than the complications could give information which operation should be preferred.
Comments on the Quality of English Language
There are a lot of typing errors throughout the text and the figures.